# Association between executive functions and COMT Val$^{108/158}$Met polymorphism among healthy younger and older adults: A preliminary study

**Zoltan Apa**[1,2]☯, **Jessica Gilsoul**[1,2]☯, **Vinciane Dideberg**[3], **Fabienne Collette**[1,2]*

**1** GIGA-CRC In Vivo Imaging, Université de Liège, Liège, Belgique, **2** Psychology and Neuroscience of Cognition Research Unit, Université de Liège, Liège, Belgique, **3** Department of Medical Genetics, University Hospital, Liège, Belgium

☯ These authors contributed equally to this work.
* f.collette@uliege.be

**Data Availability Statement:** The dataset is available from the University of Liège Institutional Data Access (https://doi.org/10.58119/ULG/DGKHVL).

## Abstract

### Background and objectives

Genetic variability in the dopaminergic system could contribute to age-related impairments in executive control. In this study, we examined whether genetic polymorphism for catechol-O-methyltransferase (COMT Val$^{158}$Met) is related to performance on updating, shifting and inhibition tasks.

### Methods

We administered a battery of executive tasks assessing updating, shifting and inhibition functions to 45 older and 55 younger healthy participants, and created composite z-scores associated to each function. Six groups were created based on genetic alleles (Val/Val, Val/Met, Met/Met) derived from the *COMT* gene and age (younger, older). Age and genotype effects were assessed with *t*-test and ANOVA (p<0.05).

### Results

A lower performance was observed in the older group for the three executive processes, and more particularly for inhibition. Moreover, older participants homozygous for the Val allele have a lower performance on the inhibition composite in comparison to younger Val/Val.

### Conclusions

These results confirm presence of executive performance decrease in healthy aging. With regard to genetic effect, older participants seem particularly disadvantaged when they have a lower baseline dopamine level (i.e., Val/Val homozygous) that is magnified by aging, and when the executive measure emphasize the need of stable representations (as in inhibition task requiring to maintain active the instruction to not perform an automated process).

**Funding:** This work was supported by the University of Liège and grant EOS 30446199 from the Belgian National Fund for Scientific Research F. R.S-FNRS. JG and ZA are research fellows (FNRS and University of Liège, respectively) and FC is Research Director at the F.R.S-FNRS.

**Competing interests:** The authors have declared that no competing interests exist.

## 1. Introduction

Aging is associated with subtle decline in cognitive functioning even in the absence of brain pathology hallmark. However, the different functions do not undergo a general decline and changes are primarily observed for high-level processes, such as in executive tasks [1,2]. Age related executive difficulties were first observed in complex and multicompound tasks involving planning, conceptualization and problem solving (for a review, see [3]). For example, lower performance by comparison to younger people is frequently reported for the three distinct functions of inhibition, shifting and updating of information identified by Miyake et al. [4] as separate cognitive processes. However, shifting and inhibition tasks are not found systematically impaired during aging [3]. For the shifting process (that also refers to flexibility abilities), the capacity to maintain and manipulate two mental sets simultaneously is more sensitive to the age-effect than the capacity to alternate between the two sets [5,6]. In the inhibition domain, there is evidence in favor of the dissociation between automatic and controlled inhibitory processes, with only the latter being sensitive to age [7–10].

Several authors proposed that changes in prefrontal cortex (PFC) structure and functions explain a large part of changes in executive functioning, [11,12]. For example, specific and global executive scores were notably found related to decreased grey matter volume in anterior brain areas [13,14], and lower executive performance was associated to decreased brain activity in prefrontal areas [15]. Presence of increased prefrontal brain activity and recruitment of bilateral regions was associated with attempts for implementation of compensatory processes (for reviews, [11,16]).

Cognitive performance during aging is also affected by diverse neurobiological factors such as neurotransmitters pathways degeneration and genetic predispositions [17,18]. Specifically, the decrease of dopamine concentration in the prefrontal cortex, [19,20] may be associated with changes in executive functioning. Catechol-O-methyltransferase (COMT) is the major enzyme involved in the degradation of released dopamine and accounts for more than 60% of the metabolic degradation of dopamine in the frontal cortex [21]. A growing body of literature reports the importance of Dopamine (DA) in human cognition [22,23], and more particularly executive functions [24]. Several authors [25–28] postulate a nonlinear relationship, namely an inverted U-shaped relationship, linking cognitive performance—notably performance to working memory (WM) tasks—with levels of cerebral dopamine. Accordingly, the hypothesis suggests that when the dopamine level is too high or too low, cognitive efficiency will decrease. The metabolic activity of the COMT enzyme is affected by polymorphism on chromosome 22q11 [29–31], resulting in Val/Val, Val/Met and Met/Met genotypes. The COMT enzyme containing Met[158] is significantly less active than the Val[158] enzyme, potentially resulting in a greater synaptic dopamine level [32,33]. As a consequence, individuals homozygous for Val allele (Val/Val) are those with the lowest level of cerebral synaptic dopamine, whereas homozygous for Met allele (Met/Met) have the highest level. Heterozygote (Val/Met) carriers present with an intermediate level of COMT activity [34].

Studies that investigated the effect of COMT polymorphism on executive functioning in younger adults reported discrepant results. Several studies observed, as expected, a better performance in the Met allele carriers population [35–38] while others reported a lack of association between COMT polymorphism and executive functioning, [39–43] or even a deleterious effect of the Met allele on executive and working memory performance [44–47].

Due to the decrease of dopamine availability in prefrontal cortex in aging [48], a positive effect of the homozygous Met/Met polymorphism (with the lower enzymatic activity and thus increased level of DA availability) may be expected. However, studies using the COMT Val[158-]Met polymorphism to understand the dopaminergic modulation of age-related effects on

working memory and executive functioning performance also led to divergent results. Some authors proposed that aging could magnify the Met allele advantage observed in younger adults [49–51]. On the other hand, other studies did not observe such a positive effect of carrying the Met allele in elderly [52–55] or even showed an advantage of carrying the Val allele [56–58]. However, the tasks administered vary in the involvement of executive processes, and one possible explanation to discrepancies observed is that not all executive processes are sensitive to COMT genetic polymorphism.

Consequently, the aim of this study was to investigate the potential influence of COMT genetic polymorphism on executive functioning through a battery of tasks, and to determine if allelic group mitigate the effect of aging in a general or specific way. To obtain measures of executive functioning relatively independent of non-executive processes, several tasks were administered for each function and composite scores specific to updating, shifting and inhibition functions were created. We first tested the effect of age on executive functioning by comparing the performance of younger and older participants on each composite score. Second, we tested the COMT allelic variance effects on the executive processes for which an age-effect was found. This was done on the six groups of subjects by combining age (younger, older) and genotype (Met/Met, Val/Met, Val/Val). At the age-group effect, we expected a higher overall performance among younger participants when compared to the older participants. For genetic effects, as Met phenotype carriers have a higher level of dopamine availability in prefrontal cortex [34], we expected they express a higher performance than the two other groups, and that age effect is dampened in that group.

## 2. Methods

### 2.1. Participants

Fifty-five younger (F/M 27/28; Mean age = 23.47 years old; SD = 3.12; range = 18–30) and 45 older (F/M 26/19; Mean age = 67.37 years old; SD = 4.9; range = 60–75) subjects were included in the present study. Participants were recruited from a local database at the GIGA-CRC. They were initially enrolled through advertisement in senior centers, universities, and drugstores. Possible subjects offered inclusion criteria presented at the laboratory first for DNA sampling using Isohelix® buccal swabs. For this study, all participants were Caucasian and native French-speaking, had normal or corrected-to-normal vision and hearing. They were screened for the following exclusion criteria: (1) neurological, psychological, or psychiatric disorders; (2) abusive consumption of alcohol or drugs; (3) diagnosis of neurodegenerative disease or dementia. The cognitive status of older participants was assessed using the Mattis Dementia Rating Scale (DRS) [59]. All older participants scored above 129, which constituted the cut-off threshold for risk of dementia [60], and scores ranged between 134–144. Sociocultural level was assessed by asking the number of years completed at school. Participants were finally ascribed to one of the six possible groups (2 age groups [younger vs older] * 3 genotypes [Val/ Val, Val/Met, Met/Met] according to their COMT Val[158]Met genotype and were invited for the behavioral assessment. The selection of participants from an existing database made it possible to form groups of roughly equivalent size for each genotype (see [61] for a similar procedure).

The study was approved by the Ethic Committee of the University Hospital in Liège (approval Belgian number B707201110947). Each participant signed informed consent form before starting the study. The study was conducted in accordance with the ethical standards described in the Declaration of Helsinki (1964). The testing session lasted between 1h30 and 2h, with short breaks provided between cognitive tasks, when required by the participant.

## 2.2. Genotyping

Genetic DNA was collected using SK-2 Isohelix® buccal swabs (Ishohelix, Cell Projects Ltd, Kent, UK). The DDK-50 DNA isolation kits were used to stabilize (during the week of the sample collection) and extract DNA swabs. The concentration of the extracted DNA was quantified using a NanoDrop 1000® or a NanoDrop Lite® spectrophotometers (Thermo Fisher Scientific, Delaware, USA). TaqMan® polymerase-chain reactions (PCR) were performed in a 96 well reaction plate containing 25 µl reaction mixture including 2 µl of DNA sample, 12.5 µl of TaqMan® genotyping master mix (Life technologies; Thermo Fisher Scientific, Darmstadt, Germany), 1.25 µl of drug metabolism genotyping assay (rs4680; Applied Biosystems; thermos Fisher Scientific, Delaware, USA) and 9.25 µl of DNA free water. Allelic discrimination was performed using fluorescence endpoint genotyping method by means of the LightCycler® 480 Instrument (Software version 1.5, Roche Diagnostics Belgium, Vilvoorde, BE). The amplification protocol encompassed a denaturation cycle of 10 minutes at 95˚C, followed by 40 cycles of amplification including 15 seconds at 92˚C and 1 minute at 60˚C. Genotyping analyses were performed at the Human Molecular Genetics department of the University Hospital (CHU) of Liège.

## 2.3. Global cognition assessment

A short battery of tasks was administered to assess that the three genotype groups did not differ on global measures of cognition, and to control for such differences in our main analyses if needed. The Raven matrices [62] and the Mill-Hill vocabulary task [63] were administered to evaluate fluid intelligence and crystallized intelligence, respectively. We also measured processing speed with the XO comparison task [64] and, for older participants, global cognitive functioning was assessed with the Mattis dementia rating scale [59], a screening tool for age-related cognitive impairment. The Mattis scale was administered before the executive tasks and the tasks assessing intelligence and processing speed were administered at the end of the testing session.

## 2.4. Executive functions assessment

A battery of six neuropsychological tests was administrated to assess the executive processes of updating, shifting and inhibition. Of the obtained scores several indices were derived to calculate composite scores that reflect each process solely and independently of the idiosyncratic characteristics of each task. Task administration was in the same order as presented below, and similar for all participants.

**2.4.1 Stroop test.** This version of the standard Stroop test [65] used the word subtest (reading color words printed in black and white), the color subtest (naming the color of crosses [XXX], and the color-word interference subtest in which participants had to name the color of the printed color and not read the word. In each subtest, participants were required to name colors aloud as quickly as possible, and errors had to be orally corrected. Time allotted for each subtask was 45 seconds. In each condition, the number of correct responses was recorded. The interference index was calculated for each participant as follow: [(color-naming condition score—color-naming word-color score) / color-naming condition score], with a low score indicating good inhibition abilities.

**2.4.2 2-Back test [66].** In this test, widely used to assess updating capacity in working memory, a list of 30 letters was presented orally. For each letter, participants had to decide whether it was identical to the one presented two steps back in the sequence. They gave their answers orally and the number of correct responses constituted the updating measure score.

**2.4.3 Plus Minus test [67].** This task was administered to evaluate shifting abilities. This test comprises three distinctive subtests. In the first subtest (A), subjects were instructed to add three to each number, whereas during the second, they had to subtract three from each number. In the third subtest, participants were instructed to alternate between adding and subtracting three to each presented number. Participants were instructed to complete the task as quickly and accurately as possible. For each list, they had to note their answers and the time to complete each list was recorded. The measure used was the shifting cost, computed with the following formula: Time subtest C–((Time subtest A + Time subtest B) / 2). Higher scores denote lower shifting capacity.

**2.4.4 Verbal fluency.** Phonemic (P, R, V) and category (animals, fruits, furniture) fluency tasks were administered [68]. Participants were instructed to produce as many words as possible, beginning with the target letter or belonging to the semantic category in 60 second for each letter and category. Measure of performance was the sum of words produced in the phonetic and semantic tasks, excluding repetitions and incorrect responses. Verbal fluency performance is usually considered as a measure of shifting abilities [69]. Indeed, fluency tasks require to implement and alternate between several search strategies. However, the contribution of search strategy processes may differ between phonological and semantic tasks [70,71], and consequently the shifting score used included all verbal fluency tasks.

**2.4.5 Wisconsin Card Sorting Test, WCST.** In the classic version of this multi-compound executive task, participants matched the stimulus card to one of the key cards according to periodically changing rule (color, shape, number of items). Revealing sorting rule was inferred by the examiner's verbal feedback (correct or incorrect) given to the subject. Several scores were computed based on the correct responses and errors made by the participant. We defined a flexibility score based on the percentage of perseverative errors from one category to the next one [4] and an inhibitory score corresponding to no-perseverative errors [72]. Higher scores are indicative of lower shifting and inhibition abilities.

**2.4.6 Random Number Generation (RNG) task.** This is a widely used and complex task allowing to assess several executive abilities [4,73,74]. In the version we used [75], participants were asked to generate a random sequence of digits between 1 and 9, at a response pace of one second indicated by a metronome. Randomness was explained with the analogy of "picking digits out of a hat and putting them back after reading". We emphasize that a random sequence would not contain a preponderance of repetitions or adjacent number values. The task was considered completed when the participant had achieved 100 correct answers or had achieved 110 correct answers in case of one or more errors. Flexibility and inhibition indices were computed using the RGCalc Quantifying order in response sequences Version1 software [75].

The *Turning Point Index (TPI)* corresponds to the number of responses that, as numerical value, correspond to a change between ascending and descending sequences. This value is compared to a theoretical random distribution over a 100-digit sequence. A higher score denoted the subject's ability to achieve a higher random sequence, and to inhibit stereotyped responses. The *Adjacency score* corresponds to the number of ascending [4,5] or descending [7,8] number pairs by comparison to the total number of response pairs. As ascending and descending pairs represent typical and relatively automatic responses, a low adjacency score is indicative of good inhibition capacity. The *runs Index* corresponds to the variability of phase length, namely the variability in the number of items between two "turning points". Based on the previous framework proposed by Miyake et al. [4], the inhibitory score was computed using the formula: (TPI–Adjutancy + Runs) / 3.

Additionally, three indices associated with updating of information were calculated. The *Redundancy Index (RI)* reflects the frequency at which each response alternative is produced.

As the selection frequency among alternatives deviates from equality, the sequence can be said to have more redundancy. The *Coupon* score indicates (across the entire set) the mean number of responses produced before all the response alternatives are given, with a range between 9 and 100, meaning that the participant has not used up all alternatives. Higher score indicates lower updating performance. Third, the *Mean Repetition Gap (MRG)* Index corresponds to the mean number of responses between two productions of the same digit. A higher score represents better updating capacity. Consequently, at individual level, low redundancy and Coupon index, and high MRG score express good updating capacities. Following the previous framework proposed by Miyake et al. [4], the updating score was calculated as (- redundancy– Coupon + Mean RG)/3.

## 2.5. Executive composite scores

The development of the executive composite scores is based on the proposal of Miyake et al. [4] that statistically demonstrated the separability of updating, shifting and inhibition processes. Specific standardized scores were created to assess the effects of age and genotype. The procedure common to the two analyses is three-steps. (1) We first standardized the eight measures extracted from the six executive tasks. (2) A mean composite score corresponding to each executive process was created by averaging the corresponding Z-scores [mean inhibition: ("Z RNG inhibition"–"Z stroop interference"–"Z WCST no-perseverative errors)/3; mean shifting: (-Z" WCST no-perseverative errors–"Z plus-minus" + "Z total fluency")/3; mean updating: ("Z correct responses 2-Back" + "Z RNG update score")/2]. The minus sign was used when a low score corresponds to a high-level performance. So final values associated to each composite correspond to *higher value = best performance*. (3) The three mean composites scores were again Z-scored. These final scores were used in statistical analyses.

For assessment of age-related effects, at step (1), data from the older group were Z-scored based on the mean and standard deviation of the younger group. At step (3), the three mean composites scores of the older participants were again Z-scored by reference to the younger group. Consequently, final composites scores in the younger group had a mean of 0 and standard deviation of 1, and we assessed deviation of performance in the older group by comparison to the younger one [76]. The same rationale was used to assess genotype effects, except that at steps (1) and (3) the Z scores were created based on the whole sample, and we assessed deviations in each of the six subgroups.

## 2.6. Statistical analyses

Statistical analyses were conducted by using JASP (Version 0.11.1.0; JASP Team, 2020). Results with $p < .05$ were considered as statistically significant. Analyses were run on the sample size of 100 participants. Missing data (four for the whole dataset) were replaced by the mean value in the participant group for this variable.

With regard to demographic data (Sex, education) and global measures of cognition (fluid and crystallized intelligence, speed of processing, Mattis scale), the effect of age was assessed with Student *t* test between younger and older participants for education level and cognitive performance. Moreover, analyses of variance (ANOVA) were performed separately in younger and older participants to determine effect of COMT polymorphism on these variables in each group. A Pearson chi-square ($\chi^2$) test of independence assessed Sex distribution across age groups and, for each age group, across the three genotypes.

The effect of age on executive functioning was assessed with Student *t* test between younger and older participants. As detailed in the method section, measures were composite Z scores centered on mean and standard deviation of the younger group. To determine if age-related

changes in executive functioning is modulated by COMT polymorphism, we performed an analysis of variance (ANOVA) with the 6 groups as an independent variable, and executive scores as an intra-subject measure. The dependent variables were composite Z-scores centered on mean and standard deviation on the whole sample. Bonferroni post-hoc tests were carried out to explore, for each executive process, the presence of difference between younger and older participants from a same genotype group or between the three genotypes within the same age group. If another comparison was statistically significant, it was reported in the result section for sake of completeness only. Data distribution and variance homogeneity for composite scores used in Student $t$ test and ANOVA were analyzed with Shapiro-Wilk and Levene tests, respectively.

A sensitivity analysis was performed for our ANOVA including the three executive composite scores. Regarding the between effect (Group effect), with a total sample size of 100 and six groups, we were able to detect effect size of $f = .30$ which corresponds to an $\eta^2 = .08$ and is an intermediate effect size ($d$ Cohen = 0.60).

## 3. Results

### 3.1. Demographic data

Demographic data associated to age group are summarized in Table 1. A Pearson chi-square ($\chi^2$) test of independence showed that the Sex distribution did not significantly differ between the younger and older subgroups [Pearson $\chi(5)^2 = 1.45$, $p = .92$]. The two groups also did not differ on education level [$t(98) = 0.11$, p = 0.91]. However, older adults had a higher vocabulary level on the French adaptation of the Mill Hill test [63] [$t(98) = -6.28$, p < .001], and younger adults better performance on Raven' matrices [$t(98) = 7.38$, p < .001] and processing speed [$t(98) = 10.45$, p < .001] tasks.

Demographic data associated to genetic polymorphism for younger and older subjects are presented in Table 2. Pearson chi-square ($\chi^2$) test of independence showed that the Sex distribution did not significantly differ between the three genotypes for younger (Pearson $\chi(2)^2 = 0.99$, $p = .61$) and older adults (Pearson $\chi(2)^2 = 1.34$, $p = .51$). ANOVAs on educational level [$F(2,52) = 0.22$], fluid intelligence [$F(2,52) = 0.89$], vocabulary level [$F(2,52) = 2.72$] and speed of processing [$F(2,52) = 1.75$] did not show significant between-group differences in younger

**Table 1. Demographic data.**

| | Younger (N = 55) | Older (N = 45) |
|---|---|---|
| Sex (F/M) | 27/28 | 26/19 |
| Genotype (Val/Val, Val/Met, Met/Met) | 21/17/17 | 15/12/18 |
| Age [1] | 23.49 (3.2) | 67.33 (4.9) |
| Education (years completed) | 14.55 (1.87) | 14.49 (3.15) |
| Raven's matrices task [2] * | 84.3 (9.64) | 68.15 (12.25) |
| Mill Hill vocabulary task [2] * | 70.75 (10.04) | 83.79 (10.67) |
| Processing Speed task [2] * | 69.06 (11.07) | 48. 35 (8.11) |
| Mattis Dementia Rating Scale [3] | / | 140.71 (2.67) |

Values are expressed as mean (standard deviations), except for Sex and Genotype.

[1] Median age is 23 for young and 67 for older

[2] percentage of correct responses

[3] Total score /144

*: p<0.001.

**Table 2. Demographic data for the three genotype groups, in younger (A) and older (B) adults.**

| | Younger participants | | |
|---|---|---|---|
| | Val/Val N = 21 | Val/Met N = 17 | Met/Met N = 17 |
| Sex (M/F) | 12/9 | 7/10 | 8/9 |
| Education (years completed) | 14.76 (2.07) | 14.41 (2.03) | 14.41 (1.5) |
| Raven's matrices task [1] | 84.52 (9) | 81.96 (10.45) | 86.37 (9.61) |
| Mill Hill vocabulary task [1] | 71.57 (11.17) | 66.44 (10.35) | 74.05 (6.68) |
| Processing Speed task [1] | 66.23 (10.88) | 68.74 (11.36) | 72.88 (10.51) |
| | Older participants | | |
| | Val/Val N = 15 | Val/Met N = 12 | Met/Met N = 18 |
| Sex (M/F) | 7/8 | 7/5 | 12/6 |
| Education | 14 (2.39) | 15.33 (3.52) | 14.33 (3.48) |
| Raven's matrices task [1] | 65.22 (12.86) | 65.97 (11.86) | 72.04 (11.6) |
| Mill Hill vocabulary task [1] | 79.8 (13) | 85.54 (9.83) | 85.95 (8.47) |
| Processing Speed task [1] * | 49.75 (7.17) | 42.59 (7.57) | 51.03 (7.63) |
| Mattis Dementia Rating Scale [2] | 141.4 (1.76) | 139.92 (2.94) | 140.67 (3.07) |

Values are expressed as mean (standard deviations), except for Sex and Genotype.

[1] percentage of correct responses

[2] Total score /144

*: p = 0.01.

participants (all p>0.05). For older adults, a lower performance for the processing speed task was observed for Val/Met participants [$F_{(2,42)} = 5$; p = 0.01, $\eta^2 = 0.19$, Bonferroni post-hoc test] while there is no significant group differences (all p>0.05) for educational level [$F_{(2,42)} = 0.63$], fluid intelligence [$F_{(2,42)} = 1.56$], vocabulary level [$F_{(2,42)} = 1.62$] and score at the Mattis dementia rating scale [$F_{(2,42)} = 1.04$].

## 3.2. Age-related effects on executive functioning

The specific standardized scores created to assess age-related effects are graphically represented in Fig 1. As described in the methods section, the final composites scores in the younger group had a mean of 0 and standard deviation of 1, and we assessed deviation of performance in the older group by comparison to the younger one [76]. Results indicated lower performance in the older group for the three executive scores [Inhibition: $t_{(98)} = 6.2$, p<0.001, $\eta^2 = 0.28$; Updating: $t_{(98)} = 2.45$, p<0.05, $\eta^2 = 0.06$; Shifting: $t_{(98)} = 2.00$, p<0.05, $\eta^2 = 0.04$], with the larger effect size for inhibition [77]. Scores [mean (standard deviation)] for older are -1.42 (1.28) for inhibition, -0.49 (1.01) for updating and -0.44 (1.2) for shifting.

As the inhibition variable was not normally distributed and we observed slight inhomogeneity of variance for updating, the analyses were also done with Satterthwaite $t$ and Mann-Withney U, respectively. The analyses confirmed between group differences. Inhibition: $t$Satterthwaite (82.06) = 6.05, p<0.001; Updating: U = 1578.5, p<0.05.

## 3.3. Genetic-related effects on executive functioning

To determine if the aging effect previously observed on executive functioning is modulated by COMT polymorphism, we performed a repeated measure analysis of variance (ANOVA) with group (Younger-Met/Met, Younger-Val/Met, Younger-Val/Val, Older-Met/Met, Older Val/

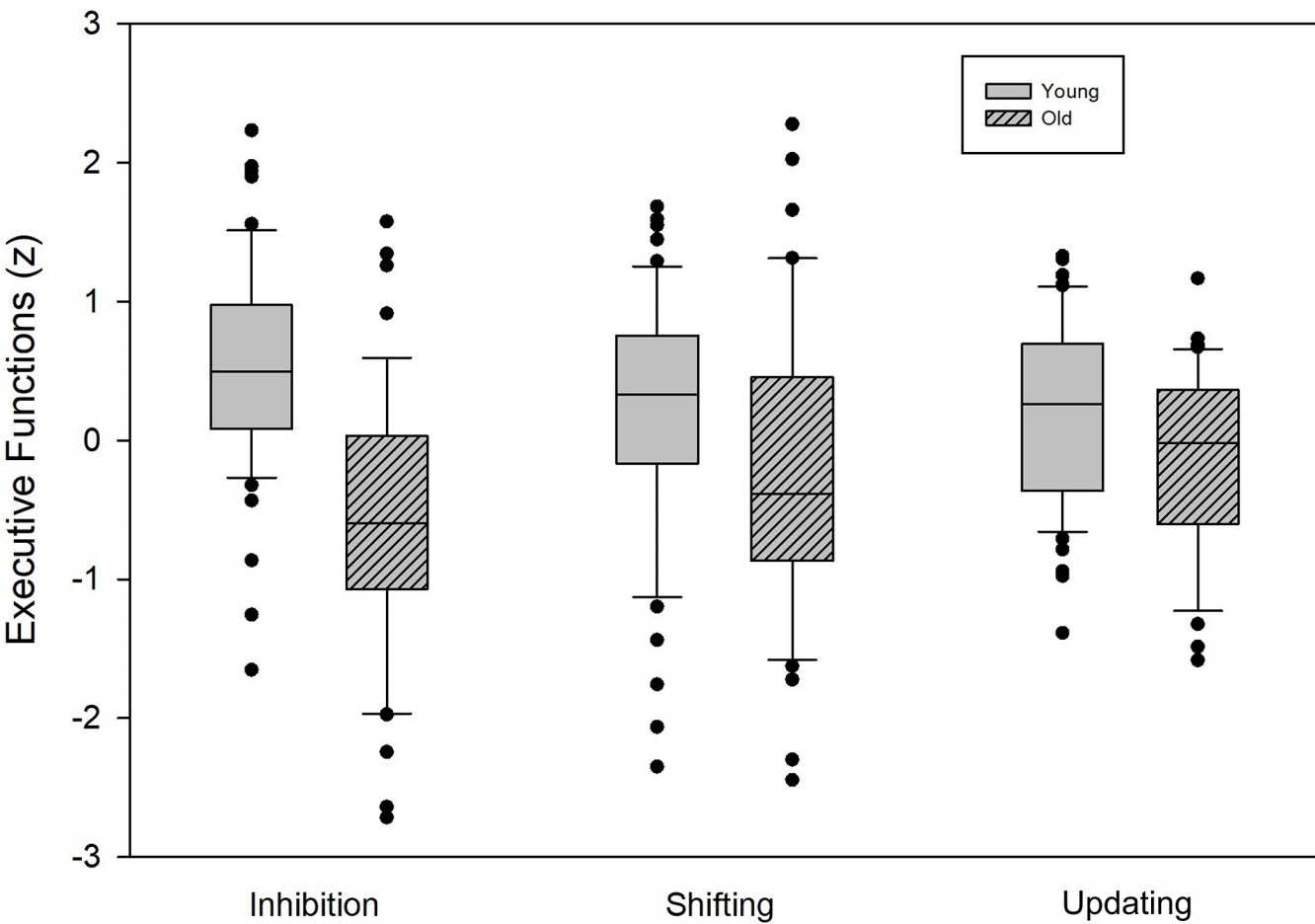

**Fig 1. Age-related effects on the three executive scores.** Composite Z-scores for the inhibition, shifting and updating functions. Scores of older participants were Z-scored by reference to the younger group, with higher values corresponding to better performance. Box plots show the median (horizontal line) and the interquartile (25–75%) range, and the whiskers indicate the 8 to 92% range. Dots indicate outlier participants.

Met, Older -Val/Val) as an independent variable, and executive scores (Inhibition, shifting, Updating) as an intra-subject measure. Results of assumptions check showed that the repeated measures ANOVA can be applied because the data did not severely violate the conditions of application. Shapiro-Wilk tests showed a normal distribution of the data [Inhibition W = 0.984 p = 0.253; Flexibility W = 0.988 p = 0.528; Updating W = 0.976 p = 0.061]. Levene's test indicates that variance are approximatively equal across the six groups [inhibition $F_{(5,94)}$ = 1.26, p = 0.29; Flexibility $F_{(5,94)}$ = 1.47, p = 0.21; Updating $F_{(5,94)}$ = 0.44, p = 0.82]. The Mauchly test revealed that sphericity was not respected $\chi^2(5)$ = 12.86, p<0.005). However, the epsilon values of the Greenhouse (0.886) and Huynh-Feldt (0.949) correction being close to 1 suggest a slight violation of sphericity.

As described in the methods section, the executive indices are the composite scores values based on the means and standard deviations of the whole sample. Results show a main group effect [$F_{(5,94)}$ = 4.3, p<0.001, $\eta^2$ = 0.1], corresponding to a large effect size [77], with a higher performance in younger participants. There is no significant effect of the executive factor [$F_{(2,188)}$ = 0.11, p = 0.9, $\eta^2$ = 0.001] and no interaction between group and executive scores factors [$F_{(10,188)}$ = 1.46, p = 0.16, $\eta^2$ = 0.07]. The results are graphically presented in **Fig 2** (see S1 Table for summary data).

Figure 2

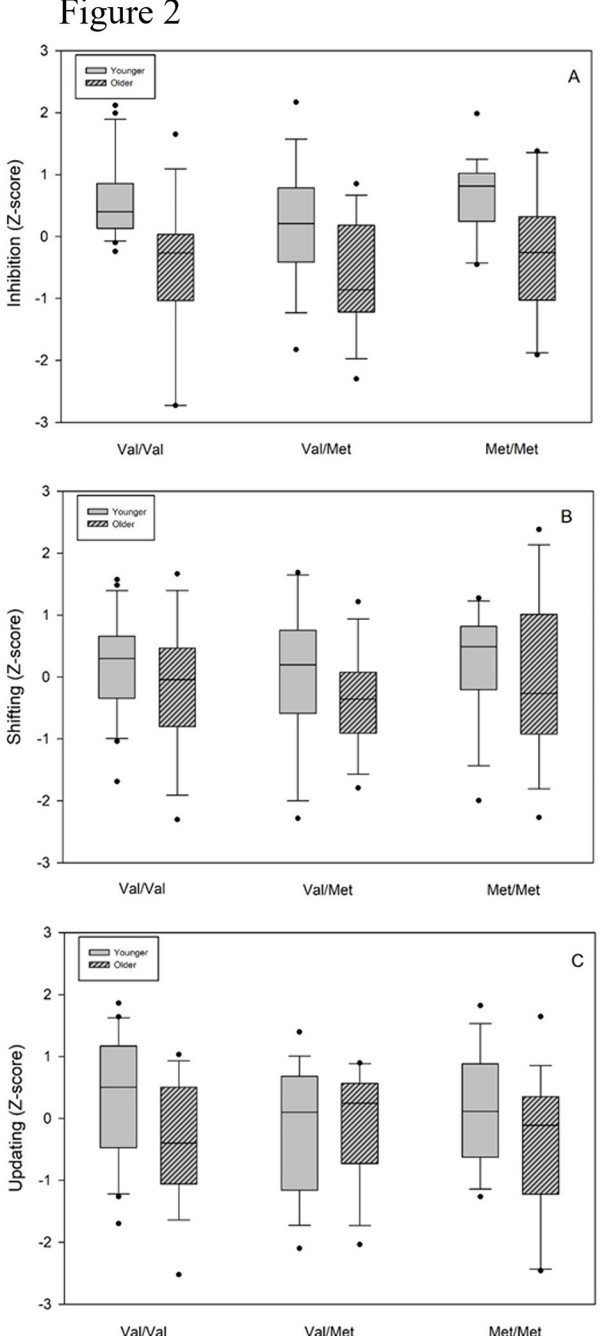

**Fig 2. Age-related effect on executive scores according to COMT polymorphism.** (A) Inhibition, (B) Shifting, (C) Updating. Z scores were created based on the whole sample, with higher values indicating better performance. Box plots show the median (horizontal line) and the interquartile (25–75%) range, and the whiskers indicate the 8 to 92% range. Dots indicate outlier participants.

As we were particularly interested by the age-effect in each genotype group for each executive process separately, Bonferroni post-hoc tests were carried out on the interaction effect. These analyses indicated that Younger Val/Val participants have a higher performance than Older Val/Val on the inhibition composite z-score (p = 0.03).

Significant differences across genotype and age groups were also observed between Younger Val/Val and Older Val/Met (p = 0.04), Younger Met/Met and older Val/Val (p = 0.04), Younger Met/Met and Older Val/Met (p = 0.04). These results are reported only for sake of completeness as we do not have specific hypotheses for these differences.

## 4. Discussion

Our study aimed to investigate possible influence of the single nucleotide polymorphism of the Val[158]Met COMT gene on executive functions during normal aging. We first replicated previous studies that showed decreased executive functioning in aging. Regarding the genetic effects, we have not observed advantage of carrying Met allele in younger, while there exists age-related effect for Val homozygous individuals on inhibition, with a lower performance for older adults by comparison to younger ones.

### 4.1. Aging and executive functions

It is now well established that aging is associated to a decrease of executive functioning; [3,78]. Here we assessed the functions of updating, shifting and inhibition with several tasks and we computed z-scores grouping together measures from the different tasks associated to the same executive process. This allows us a measurement of executive functioning relatively independent of non-executive processes [79–81]. In that way, we observed that the effect of age encompasses the three executive processes explored, with large effect size for inhibition, while shifting and updating processes are associated to small and medium effect, respectively. A decrease in performance affecting different aspects of executive functioning simultaneously has been reported by [82]. The presence of a large age-related effect size for inhibition can seem surprising as not all inhibitory processes does seem to be impaired with aging [83]. However, the tasks used here necessitate controlled processes (i.e., to actively suppress a dominant response or routine), and not automatic processes that are generally not affected by aging.

### 4.2. COMT and executive functions

At this time, few genome-wide association studies (GWAs) were conducted to assess the potential genetic contribution to executive functioning. Dueker et al. [84] observed that a composite measure of executive function is associated with four SNPs located in the long intergenic non-protein coding RNA 1362 gene, *LINC01362*, on chromosome 1. The associated SNPs have been shown to influence expression of the tubulin tyrosine ligase like 7 gene, *TTLL7* and the protein kinase CAMP-activated catalytic subunit beta gene, *PRKACB*, in several regions of the brain involved in executive function. Hatoum et al. [85] observed that genes related to GABAergic pathways influence executive functions (assessed by a common factor) while the effect of COMT Val/Met polymorphism was not significant at the genome-wide level.

We predicted a higher performance in carriers of Met allele, due to their higher level of dopamine availability in prefrontal cortex. However, we do not observe effect of COMT genotype between the three younger groups, neither for the three older groups. The advantage of carrying Met allele was frequently related to superior performance on a number of executive tasks, both in younger and older adults, as well as in pathological populations such as schizophrenia [35,86–88], while this is not systematic (for a metanalysis see [39]). There exists evidence that COMT polymorphism alone is not sufficient to explain genetic effects on executive performance. For example, Bertolino et al. [89] reported that both COMT and DAT genotypes have an additive effect on brain activity during an updating 2-back task, with COMT Met/Met and DAT 10/10-repeat individuals having a more focused cortical brain response (lesser brain activity for similar behavioral performance), whereas COMT Val and DAT 9-repeat alleles

have the least focused brain response. In our lab, we assessed the effect of COMT rs4680, MAOA rs6323 and DRD1 rs4532 polymorphism on inhibition abilities. We observed an interactive effect of the DRD1 and COMT polymorphisms on a motor inhibitory task, and an interactive effect of the three genes on a Stroop task, according to requirement of flexibility associated to the tasks [90]. Finally, in a review of the literature, Barnes et al. [91] suggest that variants of the dopamine transporter gene (DAT1) and dopamine D4 receptor gene (DRD4) may explain individual differences in both behavioral and neural measures of inhibitory control. As our protocol was restricted to Val[158]Met COMT polymorphism, we could have missed an effect of dopaminergic genes in our groups of younger and older participants.

## 4.3. COMT, aging and executive functions

Our initial prediction was a dampening of the age-effect on executive functioning in the Met/ Met group, due to their higher baseline dopamine level in prefrontal cortex [34] that should help to compensate decrease in dopamine availability associated to aging [20]. However, we observed an effect of COMT polymorphism only for older Val homozygous, by comparison to their younger counterpart. This suggests lower executive abilities in older Val/Val participants, although we were unable to identify any difference between the three genotypic groups including older participants, as previously reported by Nagel et al. [49]. These authors proposed that aging would cause individuals to migrate to the left side of the inverted U-shaped curve linking dopamine levels to cognitive performance [92,93], and Val/Val individuals would be characterized by a greater shift to the left than Met/Met ones. This greater shift of Val/Val individuals could explain the difference in cognitive performance observed here between younger Val/Val and older Val/Val subjects, whereas no significant difference was observed between younger and older Met/Met. However, this shift may be insufficient in our participants to generate a difference between the older genotype groups.

Capitalizing on previous studies, we tentatively propose that age of our participants and the use of only one genetic polymorphism can be partly responsible for the negative result we observed. Indeed, some data suggest that age-related genetic effects on cognition are observed in relatively younger participants. Two studies that assessed longitudinal decline in cognition reported an effect of COMT polymorphism only in participants younger than 60 [93,94]. Our participants are 60–75 years old, and therefore may have been outside the age range to detect a deleterious influence of the Val allele in aging if such an influence is specific to the 5th decade of life [94]. Moreover, it is also possible that the effect of COMT genotype in aging is expressed only in interaction with other genes not necessarily linked to the dopaminergic system. For example, Nagel et al. [49] observed that older COMT Val homozygotes have particularly low levels of performance if they were also BDNF Met carriers. Erickson et al. [95] reported that the COMT polymorphism does not affect the trajectory of age-related executive control decline, whereas the Val/Val polymorphism for BDNF may promote faster rates of cognitive decay in older age. Sapkota et al. [96] observed that additive effects of COMT and BDNF polymorphisms predicted lower executive performance. In a subsequent study, the authors also showed that APOEε4+ carriers with increased allelic risk for COMT and BDNF genes had lower composite executive score [97]. Finally, an interactive effect was also observed for both COMT and APOE for semantic fluency, with Val/Met heterozygous individuals performing better only in the context of an ApoE ε4 allele [98].

## 4.4. COMT, aging and inhibition

We observed an age-related effect for Val homozygous carriers only for inhibition abilities. One possible interpretation is in line with the findings of Bilder et al. ([48]; see also [99]) that

COMT polymorphism has allelic specific and opposite effects on the stability and flexibility of cognitive processes. According to these authors and our results, the *COMT* Met allele supports the stability of representations necessary for the ongoing cognitive task while the Val allele favors switching abilities and consequently updating/flexibility of these representations. Consequently, the Val carriers might be advantaged on tasks requiring flexibility and updating mechanisms and the Met carriers by tasks requiring stable maintenance abilities [48,99–101]. In other words, Val/Val individuals would have lower capacity for tasks requiring stability of representations, which underlies inhibition processes [102]. This would explain why we observed and effect of age for inhibitory abilities, but not updating nor shifting, in Val homozygous carriers. A similar interpretation was proposed by de Frias et al. [103] to explain differences in pattern of brain activity during an *2-back* task: increased transient activity in the medial temporal lobe was observed in the Met/Met group and interpreted as a decrease of updating abilities while increased sustained activity in the prefrontal cortex was reported for the Val/Val group and attributed to less efficient maintenance processes (see also [101,104]).

### 4.5. Strengths and limitations

To the best of our knowledge this is the first study to incorporate and analyze the three executive processes of inhibition, shifting and updating among younger and older participants in the context of COMT allelic variance. We demonstrated that inhibition, shifting and updating performances are distinctly affected by COMT polymorphism in each age group. Another strength of our study is the use of composite scores which enabled us to obtain a purer and more distinguishable measure of the executive processes core components, thus taking further the prominent model proposed by Miyake et al. [4].

As previously discussed, one possible limitation is the restricted age range of our older sample which does not allow us to determine whether the effects of COMT polymorphism differ in young-old and old-old participants [94]. Finally, our sample size is well in line with other behavioral studies which were aiming to investigate cognitive processes, but slightly smaller, however adequately sized, compared to the previous studies on executive functioning and COMT effect in aging [105]. Consequently, presence of a lower performance during aging on the composite score of inhibition (but not on shifting or updating scores) for homozygous Val carriers needs replication with protocols targeting for example one specific executive process. This will allow inclusion of a larger number of participants than in this study.

### 5. Conclusion

This study used a battery of tasks to explore the effect of COMT polymorphism on executive performance in aging by using compound scores that allow a purer measure of the executive processes of updating, shifting and inhibition. We showed, as expected, lower performance in the older group for the three executive processes. However, older participants seem disadvantaged only when the executive measure emphasize the need of stable representations (as in inhibition task requiring the instruction to not perform an automated process) and when they are homozygous for Val allele, with a lower baseline dopamine level that is magnified by aging. Until now, studies on the effect of COMT polymorphism on executive functioning (and more generally cognition) in aging led to divergent results, however we can conclude here on the evidence of COMT allelic variance effect on the inhibitory component of executive functioning in older participants. Further studies are needed to investigate and disentangle the effect of task characteristics, interaction with other genes (e.g., BDNF) or if the effect is specific to a very narrow age range.

## Supporting information

**S1 Table. Summary data for executive composite scores.**
(DOCX)

## Author Contributions

**Conceptualization:** Jessica Gilsoul, Fabienne Collette.

**Data curation:** Jessica Gilsoul.

**Formal analysis:** Zoltan Apa, Jessica Gilsoul, Vinciane Dideberg.

**Funding acquisition:** Fabienne Collette.

**Investigation:** Jessica Gilsoul.

**Methodology:** Vinciane Dideberg, Fabienne Collette.

**Project administration:** Fabienne Collette.

**Supervision:** Fabienne Collette.

**Validation:** Fabienne Collette.

**Writing – original draft:** Zoltan Apa, Jessica Gilsoul.

**Writing – review & editing:** Zoltan Apa, Jessica Gilsoul, Vinciane Dideberg, Fabienne Collette.

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
