## [Decision Letter · Decision Letter 0]

25 Sep 2023

PONE-D-23-21633

Association between executive functions and COMT Val108/158Met polymorphism among healthy younger and older adults

PLOS ONE

Dear Dr. Collette,

Thank you for submitting your manuscript to PLOS ONE. After careful consideration, we have decided that your manuscript does not meet our criteria for publication and must therefore be rejected.

Specifically:

I am sorry that we cannot be more positive on this occasion, but hope that you appreciate the reasons for this decision.

Kind regards,

Álvaro Astasio-Picado

Academic Editor

PLOS ONE

Reviewers' comments:

Reviewer's Responses to Questions

**Comments to the Author**

1. Is the manuscript technically sound, and do the data support the conclusions?

Reviewer #1: No

Reviewer #2: Partly

2. Has the statistical analysis been performed appropriately and rigorously? 

Reviewer #1: No

Reviewer #2: Yes

3. Have the authors made all data underlying the findings in their manuscript fully available?

Reviewer #1: No

Reviewer #2: Yes

4. Is the manuscript presented in an intelligible fashion and written in standard English?

Reviewer #1: No

Reviewer #2: No

5. Review Comments to the Author

Reviewer #1: Review PONE-D-23-21633

Association between executive functions and COMT Val108/158Met polymorphism among healthy younger and older adults

Overview

The manuscript investigated the association of executive functions and polymorphism in young and older adults. In general, the manuscript is organized. I suggest improving the Figure's quality. For example, in Figure 2, it is difficult to understand the groups.

Specific comments:

Keywords

The keywords have “COMT and Executive functions”, both in the manuscript's Title. A suggestion for keywords is not to use the same ones already in the title because it increases the chances of your work being found on search engines.

Introduction

This session introduces three executive functions (EF), i.e., inhibition, shifting and updating, based on the study of Miyake et al. (2000). This study investigated the contributions of EF to Complex “Frontal Lobe” Tasks. However, in the Introduction, it was defined the PFC area. Other EFs may deteriorate with age, such as planning, reaction time or flexibility. So, please clarify why it was used only these three EFs.

Method

The sample was described by age and males in each group (e.g., Younger: 28 Men, M age = 23.47 years old; SD = 3.12; range = 18-30). What about the women’s characteristics? Please describe it in both groups of age.

There is missing information in this session. Let me be clear: In what order were the tests administered? Was the test order controlled between participants? How long did it last? Was there an interval between the tests? Were all the tests administered in a single session or multiple sessions? Please insert all procedures to allow one to replicate the experiment.

Please insert a reference for the classes adopted for effect t size.

Please insert procedures for analyzing data normality before data analyses.

Item 2.6 Statistical analyses didn't describe the Statistical procedures, but only the software. Please describe all statistical procedures, including data normality and sample size.

Results

When an ANOVA has main effects, post hoc is carried out to identify the differences. The item 3.3. Genetic-related effects on executive functioning, ANOVA didn’t have main interactions, and there is no need to run a post-hoc. Although you can find a few papers that run post-hoc without main interaction, running Bonferroni's post-hoc when the “p” value was high, and effect size low (i.e., p=0.16, ƞ2 =0.07) is quite strange. Following this way of thinking, these results cannot be discussed.

Discussion

Since the results of both hypotheses are not considered (please, check the results comments), the discussion should go in the opposite direction and be rewritten.

Reviewer #2: Article „Association between executive functions and COMT Val108/158Met polymorphism among

healthy younger and older adults“ written by Apa et al. is impressive in the number of used neuropsychological tests. However, the number of subjects involved in this study (overall 100 participants) is too small for serious genetic analysis. Since you have healthy participants, you should include at least 500 participants. Thus, I suggest the rejection of this article. These are my further remarks:

Abstract

- it should be „in comparison to younger Val/Val.“ Instead of „by comparison to younger Val/Val.“

Methods

- you should not present age as a mean value, but as median

- what does the numeric unit „ll“ stand for (you mentioned it in the paragraph 2.2.)

- you should write „°C“ instead of“_C“

- Paragraph „2.6. Statistical analyses“ should be written more thoroughly. Authors should explain which exact tests they used in their analysis (not only the statistical programme in which the analysis was performed).

Results

- In section „Methods“ the authors used period as decimal point, while in section „Results“ authors used comma as decimal point. This has to be corrected.

- In the „Table 1.“ you should also add the column on the results of statistical analyses.

- You should not mention statistical tests in the „Results“ section, but you should explain which test you used for what kind of analysis in paragraph „2.6. Statistical analyses“.

- In the „Table 2.“ you should also add the column on the results of statistical analyses (comparison of variables between the groups).

- The resolution of figure 2. is poor and it should be improved.

Discussion

- you should mention if any GWAS studies were conducted before by using the results of cognitive tests you used in your study as quantitative traits.

- instead of „nucleotide polymorphism“ should be „single nucleotide polymorphism“

- References are poorly written, sometimes you write them as Vancouver style, and sometimes as Harvard style.

6. PLOS authors have the option to publish the peer review history of their article (what does this mean?). If published, this will include your full peer review and any attached files.

Reviewer #1: **Yes: **Herbert Ugrinowitsch

Reviewer #2: No

- - - - -

---

## [Author Response · Author response to Decision Letter 0]

27 Oct 2023

Reviewer #1

Comment 1. Overview

The manuscript investigated the association of executive functions and polymorphism in young and older adults. In general, the manuscript is organized. I suggest improving the Figure's quality. For example, in Figure 2, it is difficult to understand the groups.

Response. As suggested by Reviewer 1, quality of our two Figures was improved. 

Comment 2. Keywords

The keywords have “COMT and Executive functions”, both in the manuscript's Title. A suggestion for keywords is not to use the same ones already in the title because it increases the chances of your work being found on search engines. 

Response. Based on the reviewer’s comment, we have changed the keywords accordingly. Keywords are now: Aging, Inhibition, Shifting, Updating, Cognitive control, Genetics (Page 2) 

Comment 3. Introduction

This session introduces three executive functions (EF), i.e., inhibition, shifting and updating, based on the study of Miyake et al. (2000). This study investigated the contributions of EF to Complex “Frontal Lobe” Tasks. However, in the Introduction, it was defined the PFC area. Other EFs may deteriorate with age, such as planning, reaction time or flexibility. So, please clarify why it was used only these three EFs. 

Response. We have slightly reformulated the first two sections of the introduction to state more clearly that we do not consider that only inhibition, shifting and updating are impaired by aging, but that we are interested in these three functions because Miyake et al. (2000) showed they refers to separate cognitive constructs. We have also specified that flexibility is impaired by aging and that flexibility and shifting refers to very similar concepts. Finally, we have also specified that executive functions does not totally rely on frontal lobes.

Page 3:1:3: Age related executive difficulties were first observed in complex and multicompound tasks involving planning, conceptualization and problem solving (for a review see (3)). For example, lower performance by comparison to younger people is frequently reported for the three distinct functions of inhibition, shifting and updating of information identified by Miyake et al. (4) as separate cognitive processes. However, shifting and inhibition tasks are not found systematically impaired during aging (for a review Collette & Salmon, 2014). For the shifting process (that also refers to flexibility abilities), the capacity to maintain and manipulate two mental sets simultaneously is more sensitive to the age-effect than the capacity to alternate between the two sets (Verhaeghen & Cerella, 2002; Wasylyshyn et al., 2011 for meta-analyses).

Page 3:2:1: Several authors proposed that changes in prefrontal cortex (PFC) structure and functions explain a large part of changes in executive functioning, (for reviews, Dennis & Cabeza, 2008; Park & Reuter-Lorenz, 2009).

Comment 4. Method

(a) The sample was described by age and males in each group (e.g., Younger: 28 Men, M age = 23.47 years old; SD = 3.12; range = 18-30). What about the women’s characteristics? Please describe it in both groups of age.

Response. In the previous version of the manuscript, we reported the number of males in our sample for brevity’s sake. We have now specified the number of males/females. However, as a gender comparison will be not performed, the description of demographic information separately for both groups does not seem useful here. 

Page 6:2:1. Fifty-five younger (F/M 27/28; M age = 23.47 years old; SD = 3.12; range = 18-30) and 45 older (F/M 26/19; M age = 67.37 years old; SD = 4.9; range = 60-75) subjects were included in the present study.

(b) There is missing information in this session. Let me be clear: In what order were the tests administered? Was the test order controlled between participants? How long did it last? Was there an interval between the tests? Were all the tests administered in a single session or multiple sessions? Please insert all procedures to allow one to replicate the experiment.

Response. The information about testing session is now added.

Page 6:2:4. The testing session lasted between 1h30 and 2h, with short breaks provided between cognitive tasks, when required by the participant.

Page 7:2:4. Task administration was in the same order as presented below, and similar for all participants.

Page 10:3:7. The Mattis scale was administered before the executive tasks and the tasks assessing intelligence and processing speed were administered at the end of the testing session.

(c) Please insert a reference for the classes adopted for effect t size.

Response. We do not understand the comment of the reviewer.

(d) Please insert procedures for analyzing data normality before data analyses.

Item 2.6 Statistical analyses didn't describe the Statistical procedures, but only the software. Please describe all statistical procedures, including data normality and sample size.

Response. A lack of description of our statistical analyses was also mentioned by Reviewer 2. Statistical information was sparsely provided in the results section but we agree with the reviewers that this was not complete and explicit enough. Consequently, our analyses are now detailed at section 2.6. Results of assumptions check for student t test were already presented in the previous version of the manuscript. (footnote in the result section, Page 15:1:1 in the current version), and assumption checks are now added for the ANOVA:

Page 12:2:2. Results with p < .05 were considered as statistically significant. Analyses were run on the sample size of 100 participants. Missing data (four for the whole dataset) were replaced by the mean value in the participant group for this variable.

The effect of age on executive functioning was assessed with Student t test between younger and older participants. As detailed in the method section, measures were composite Z scores centered on mean and standard deviation of the younger group. To determine if age-related changes in executive functioning is modulated by COMT polymorphism, we performed an analysis of variance (ANOVA) with the 6 groups as an independent variable, and executive scores as an intra-subject measure. The dependent variables were composite Z-scores centered on mean and standard deviation on the whole sample. Bonferroni post-hoc tests were carried out to explore, for each executive process, the presence of difference between younger and older participants from the same genotype or between the three genotypes within the same age group. If another difference was statistically different, it was reported in the result section for sake of completeness only. Data distribution and variance homogeneity for composite scores used in Student t test and ANOVA were analyzed with Shapiro-Wilk and Levene tests, respectively. 

Page 15:3:5: Results of assumptions check showed that the repeated measures ANOVA can be applied because the data did not severely violate the conditions of application. Shapiro-Wilk tests showed a normal distribution of the data [Inhibition W=0.984 p=0.253; Flexibility W=0.988 p=0.528; Updating W=0.976 p=0.061]. Levene’s test indicates that variance are approximatively equal across the six groups [inhibition F(5,94)=1.26, p=0.29; Flexibility F(5,94)=1.47, p=0.21; Updating F(5,94)=0.44, p=0.82]. The Mauchly test revealed that sphericity was not respected χ2(5)=12.86, p<0.005). However, the epsilon values of the Greenhouse (0.886) and Huynh-Feldt (0.949) correction being close to 1 suggest a slight violation of sphericity. 

(e) Results. When an ANOVA has main effects, post hoc is carried out to identify the differences. The item 3.3. Genetic-related effects on executive functioning, ANOVA didn’t have main interactions, and there is no need to run a post-hoc. Although you can find a few papers that run post-hoc without main interaction, running Bonferroni's post-hoc when the “p” value was high, and effect size low (i.e., p=0.16, ?2 =0.07) is quite strange. Following this way of thinking, these results cannot be discussed. 

Response. We do not agree with this comment. Indeed, with a priori hypotheses, we can look at all the contrasts we consider of interest. However, in the absence of an interaction, we need to be very careful about the conclusions that can be drawn, and the results can't be interpreted as if there was an interaction. We can only say that there is a difference for one group, but not for another, without being able to state that this difference in differences is robust. We agree that our a priori hypotheses were not clearly stated in the previous version of the manuscript. This is now done Page 12:3:7. Moreover, in the previous version of the manuscript, it was already stated in the result section that unexpected significant differences are reported only for sake of completeness (Page 20:2:3 current version) and in our discussion, we refrained from discussing the post-hoc results as interaction effect.

Page 12:3:7: Bonferroni post-hoc tests were carried out to explore, for each executive process, the presence of difference between younger and older participants from a same genotype group or between the three genotypes within the same age group. If another comparison showed a significant difference, it was reported in the result section for sake of completeness only. 

Comment 5. Discussion

Since the results of both hypotheses are not considered (please, check the results comments), the discussion should go in the opposite direction and be rewritten. 

Response. We do not agree with this comment. Indeed, we also considered in the previous version of the manuscript the possibility of an absence of COMT effect due to the age of our participants and other genetic influence not controlled for in this study (see pages 18:2, 19:3, and 20:1, current version of the manuscript, no changes made). 

Reviewer #2

Comment 1. Overview

Article “Association between executive functions and COMT Val108/158Met polymorphism among healthy younger and older adults“ written by Apa et al. is impressive in the number of used neuropsychological tests. However, the number of subjects involved in this study (overall 100 participants) is too small for serious genetic analysis. Since you have healthy participants, you should include at least 500 participants. Thus, I suggest the rejection of this article. 

Response: As indicated in the Cover Letter and the 4.5 section Strength and Limitations (previous version of the manuscript), the interest of the study is the complete phenotyping of executive functioning with several measures for each function that allow computing composite scores that are purer measures of executive functioning. In that sense, the number of participants corresponds to the standard in the cognitive psychology research field. This was stated in the previous version of the manuscript (Page 24:3:3 current version). To take into account comment 1 of Reviewer 2, we have added a sentence indicating that, due to our sample size, these results need replication.

Page 22:2:6: Consequently, presence of a lower performance during aging on the composite score of inhibition (but not on shifting or updating scores) for homozygous Val carriers needs replication with protocols targeting for example one specific executive process. This will allow inclusion of a larger number of participants than in this study. 

Comment 2. Abstract

It should be „in comparison to younger Val/Val.“ Instead of „by comparison to younger Val/Val.“ 

Response: We have modified the manuscript accordingly (see Page 2).

Comment 3. Methods

(a) you should not present age as a mean value, but as median

Response. It is common practice in cognitive aging studies to report age mean along with range. For consistency with other published studies on the same topic, we prefer to keep mean and standard deviation. 

(b) what does the numeric unit „ll“ stand for (you mentioned it in the paragraph 2.2.)

- you should write „°C“ instead of“_C“

Response. We apologize for the typos remaining in that part of the manuscript. “II” stand for “µl” and “_C” for “°C”. We must thank the reviewer for the attentive reading. It allows us to rewrite all the paragraph in a more appropriate way (see Pages 6-7). 

(c) Paragraph „2.6. Statistical analyses“ should be written more thoroughly. Authors should explain which exact tests they used in their analysis (not only the statistical programme in which the analysis was performed).

Response. We thank the Reviewer for pointing out the Statistical analyses section issue. A full description has now been added to the revised version of the manuscript. A similar comment was addressed by Reviewer 1 (see Comment 4d and Page 12:2:2, 15:2:1 and 15:3:5 in the current version of the manuscript).

Comment 4. Results

(a) In section „Methods“ the authors used period as decimal point, while in section „Results“ authors used comma as decimal point. This has to be corrected.

Response: We agree with the reviewer that decimal separator should be used consistently. Hence, the decimal separator has been changed to period throughout the revised version of the manuscript. 

(b) In the „Table 1.“ you should also add the column on the results of statistical analyses.

Response: Results of statistical analyses were indicated in the text before Table 1 (section Demographic data) in the previous version of the manuscript (now Page 13). There will be redundancy if we add it also in Table 1

 (c) You should not mention statistical tests in the „Results“ section, but you should explain which test you used for what kind of analysis in paragraph „2.6. Statistical analyses“.

Response: A similar comment was addressed by Reviewer 1 and we modified the text accordingly (see Comment 4d and Page 12:3:1). However, we kept in footnotes (Page 15 and 17) rationale for confirmatory supplementary statistical analyses when normal distribution or variance homogeneity is not present.

(d) In the „Table 2.“ you should also add the column on the results of statistical analyses (comparison of variables between the groups).

Response: Results of statistical analyses are now added. For consistency with the comparison between younger and older participants, the results are included in the main text.

Page 13:2:4: ANOVAs on educational level [F(2,52)=0.22], fluid intelligence [F(2,52)=0.89], vocabulary level [F(2,52)=2.72] and speed of processing [F(2,52)=1.75] did not show significant between-group differences in younger participants (all p>0.05). For older adults, a lower performance for the processing speed task was observed for Val/Met participants [F(2,42)=5; p=0.01, ƞ² =0.19, Bonferroni post-hoc test] while there is no significant group differences (all p>0.05) for educational level [F(2,42)=0.63], fluid intelligence [F(2,42)=1.56], vocabulary level [F(2,42)=1.62] and score at the Mattis dementia rating scale [F(2,42)=1.04].

(e) The resolution of figure 2. is poor and it should be improved.

Response. The resolution of Figure 2 (as well as Figure 1) was improved. Please note that we lost resolution at the previous submission due to uploading guidelines. 

Comment 5. Discussion

(a) You should mention if any GWAS studies were conducted before by using the results of cognitive tests you used in your study as quantitative traits.

Response: GWAs studies are now included in the discussion. We capitalized on this comment to also add two recent publications in the discussion. 

Page 18:2:1: At this time, few genome-wide association studies (GWAs) were conducted to assess the potential genetic contribution to executive functioning. Dueker et al (81) observed that a composite measure of executive function is associated with four SNPs located in the long intergenic non-protein coding RNA 1362 gene, LINC01362, on chromosome 1. The associated SNPs have been shown to influence expression of the tubulin tyrosine ligase like 7 gene, TTLL7 and the protein kinase CAMP-activated catalytic subunit beta gene, PRKACB, in several regions of the brain involved in executive function. Hatoum et al. (82) observed that genes related to GABAergic pathways influence executive functions (assessed by a common factor) while the effect of COMT Val/Met polymorphism was not significant at the genome-wide l

---

## [Decision Letter · Decision Letter 1]

7 Mar 2024

PONE-D-23-21633R1Association between executive functions and COMT Val108/158Met polymorphism among healthy younger and older adultsPLOS ONE

Dear Dr. Collette,

Thank you for submitting your manuscript to PLOS ONE. After careful consideration, we feel that it has merit but does not fully meet PLOS ONE’s publication criteria as it currently stands. Therefore, we invite you to submit a revised version of the manuscript that addresses the points raised during the review process.

We look forward to receiving your revised manuscript.

Kind regards,

Kenji Tanigaki, Ph.D., M.D.

Academic Editor

PLOS ONE

Journal Requirements:

"This work was supported by the University of Liège and grant EOS 30446199 from the Belgian National Fund for Scientific Research FRS-FNRS. JG and ZA are research fellows (FNRS and University of Liège, respectively) and FC is Research Director at the FRS.-FNRS."

"This work was supported by the University of Liège and grant EOS 30446199 from the Belgian National Fund for Scientific Research FRS-FNRS. JG and ZA are research fellows and FC is Research Director at the FRS.-FNRS."

"This work was supported by the University of Liège and grant EOS 30446199 from the Belgian National Fund for Scientific Research FRS-FNRS. JG and ZA are research fellows (FNRS and University of Liège, respectively) and FC is Research Director at the FRS.-FNRS."

Additional Editor Comments (if provided):

Reviewers' comments:

Reviewer's Responses to Questions

**Comments to the Author**

1. If the authors have adequately addressed your comments raised in a previous round of review and you feel that this manuscript is now acceptable for publication, you may indicate that here to bypass the “Comments to the Author” section, enter your conflict of interest statement in the “Confidential to Editor” section, and submit your "Accept" recommendation.

Reviewer #2: (No Response)

Reviewer #3: (No Response)

2. Is the manuscript technically sound, and do the data support the conclusions?

Reviewer #2: Partly

Reviewer #3: Yes

3. Has the statistical analysis been performed appropriately and rigorously? 

Reviewer #2: Yes

Reviewer #3: No

4. Have the authors made all data underlying the findings in their manuscript fully available?

Reviewer #2: No

Reviewer #3: No

5. Is the manuscript presented in an intelligible fashion and written in standard English?

Reviewer #2: Yes

Reviewer #3: Yes

6. Review Comments to the Author

Reviewer #2: Since the authors could not include at least 500 participants in their genetic study, as I requested, they should change the article's title. The title should be; „Association between executive functions and COMT Val108/158Met polymorphism among healthy younger and older adults; a preliminary study“.

I do not agree with the authors that the practice is to present data on age as a mean value, and I think it should be presented as a median. This is in fact, a common statistical error, and you have to correct this.

Although the authors corrected the manuscript according to my other comments, I still have concerns regarding the number of the included participants.

Reviewer #3: The paper by Apa and colleagues presents a very interesting investigation on how different genotypes of the COMT gene can influence performance in various tests evaluating executive functions in both young and older individuals. The authors find a generally inferior performance in the older group across the three executive processes and, additionally, lower performance in the inhibition subprocess in older subjects homozygous for valine compared to young individuals with the same genotype. This research is interesting, but it needs improvement in several aspects to be published in Plos One.

• It would be helpful, for the review process, to include line numbers in the document.

• In the method section, information about the type of sociodemographic data collected should be included. Also, in the statistical analysis description section, it would be useful to include how these data will be analyzed.

• In the description of the N-back test, it would be clearer to simply refer to it as the 2-back test.

• In my opinion, the inclusion of a verbal fluency test as an index of shifting is not fully justified. I believe that a more detailed explanation is needed on how this test relates to that executive subprocess.

• The "Global cognition" section should appear just before the description of the neuropsychological tests used, as it represents an initial evaluation of the subjects.

• However, my major concern is about the way the data analyses were performed as they seem redundant at some point. The authors use t-tests (to analyze the results of both groups regardless of genotype) and ANOVAs (to see the effects of group and genotype) and subsequently report group comparisons from t-tests and also the main group effects in the ANOVA. This information is practically the same. In my opinion, it would only be necessary to use the ANOVA to observe these effects. Additionally, I do not quite understand why the authors have "created" 6 groups. Using a 2-way ANOVA can include group (with two levels: young and older) and genotype (three levels: Val/Val, Met/Met, and Met/Val) as factors.

• I believe it would be advisable to perform the Hardy-Weinberg equilibrium test to analyze if the genotype distributions in the sample are comparable to the general population.

• Finally, the sample size may be considered small for a genetic study; however, the authors could report statistical power or at least conduct sensitivity analyses to observe possible violations of Type II error.

7. PLOS authors have the option to publish the peer review history of their article (what does this mean?). If published, this will include your full peer review and any attached files.

Reviewer #2: No

Reviewer #3: No

---

## [Author Response · Author response to Decision Letter 1]

29 Mar 2024

We are sincerely grateful for the reviewing process. Please find enclosed the new version of the manuscript previously entitled “Association between executive functions and COMT Val108/158Met polymorphism among healthy younger and older adults”, which we have the pleasure to resubmit for publication in Plos One. We are thankful to the Reviewers for their examination of our manuscript and for providing comments that helped to improve its content. 

You will find a point-by-point reply to the topics raised by the Reviewers along with a mention of the pages and paragraph containing the changes (underlined in yellow) carried out in the new version of the manuscript.

Reviewer #2:

Comment 1. Since the authors could not include at least 500 participants in their genetic study, as I requested, they should change the article's title. The title should be; „Association between executive functions and COMT Val108/158Met polymorphism among healthy younger and older adults; a preliminary study“.

Response. This was done as requested by the reviewer #2, see Title Page 1: “Association between executive functions and COMT Val108/158Met polymorphism among healthy younger and older adults : a preliminary study”

Comment 2. I do not agree with the authors that the practice is to present data on age as a mean value, and I think it should be presented as a median. This is in fact, a common statistical error, and you have to correct this.

Response. In a review of research methods, Pickering (2017) discussed how to present descriptive statistics related to participants. According to this paper, it is prescribed to report age as mean, except in case of a skewed distribution. Regarding our data, we observed skewness values of 0.417 (younger) and 0.290 (older) that correspond to a distribution approximatively symmetric (range for normal distribution between -0.5 and 0.5. Finally, we checked if the median value corresponds to the mean value in the two groups, and we did not observe large difference (younger: 23 vs 23.49; older: 67 vs 67.33). For these reasons, we preferred to refer to mean value in the table, and added median value as footnote (Table 1, Page 13: Lines 5-7). 

Comment 3. Although the authors corrected the manuscript according to my other comments, I still have concerns regarding the number of the included participants.

Response. We acknowledge that our results would benefit from a larger number of participants. However, as discussed in the first round of revision, it was not possible to include as many participants as in genetic studies due to extensive neuropsychological testing. We hope that adding a sensitivity analysis as requested by Reviewer 2, in addition of explicit statement in the previous version that sample size is a limitation of our study, will remove at least partly your concerns.

Reviewer #3: 

The paper by Apa and colleagues presents a very interesting investigation on how different genotypes of the COMT gene can influence performance in various tests evaluating executive functions in both young and older individuals. The authors find a generally inferior performance in the older group across the three executive processes and, additionally, lower performance in the inhibition subprocess in older subjects homozygous for valine compared to young individuals with the same genotype. This research is interesting, but it needs improvement in several aspects to be published in Plos One.

Comment 1. It would be helpful, for the review process, to include line numbers in the document.

Response. This is now done as suggested by Reviewer #3

Comment 2. In the method section, information about the type of sociodemographic data collected should be included. Also, in the statistical analysis description section, it would be useful to include how these data will be analyzed.

Response. For sociodemographic data, we have now added Page 6: Lines 13-14 : “Sociocultural level was assessed by asking the number of years completed at school”. We have also added in the statistical analysis description section Page 12: lines 16-22: “With regard to demographic data (Sex, education) and global measures of cognition (fluid and crystallized intelligence, speed of processing, Mattis scale), the effect of age was assessed with Student t test between younger and older participants for education level and cognitive performance. Moreover, analyses of variance (ANOVA) were performed separately in younger and older participants to determine effect of COMT polymorphism on these variables in each group. A Pearson chi-square (χ2) test of independence assessed Sex distribution across age groups and, for each age group, across the three genotypes.”

Comment 3. In the description of the N-back test, it would be clearer to simply refer to it as the 2-back test.

Response. The N-back task we used is now referred a 2-back task across the text (see Page 8: line 19, Page 12: line 1, Page 19: line 22, Page 22: line 13)

Comment 4. In my opinion, the inclusion of a verbal fluency test as an index of shifting is not fully justified. I believe that a more detailed explanation is needed on how this test relates to that executive subprocess.

Response. As indicated in the previous version of the manuscript, the verbal fluency score was considered as a measure of shifting abilities by Eslinger & Grattan (1993). Indeed, besides assessing access to phonological and semantic information stored in long-term memory, verbal fluency tasks are usually considered as requiring strategic search within phonological and semantic networks. There is nevertheless debate on the importance of contribution of shifting abilities to performance on phonemic and semantic fluency (for example, Nutter-Hupham et al., 2008; Troyer et al., 1997). Consequently, we considered that the best measure to include in the shifting composite score was the sum of words produced for both phonemic and semantic tasks. This is now briefly state Page 9: line 14-18: “Verbal fluency performance is usually considered as a measure of shifting abilities (65). Indeed, fluency tasks requires to implement and alternate between several search strategies. However, the contribution of search strategy processes may differ between phonological and semantic tasks (70, 71), and consequently the shifting score used included all verbal fluency tasks”.

70. Troyer, A. K., Moscovitch, M., & Winocur, G. (1997). Clustering and switching as two components of verbal fluency: Evidence from younger and older healthy adults. Neuropsychology, 11(1), 138–146. doi:10.1037/0894-4105.11.1.138

71. Nutter-Upham, K. E., Saykin, A. J., Rabin, L. A., Roth, R. M., Wishart, H. A., Pare, N., & Flashman, L. A. (2008). Verbal fluency performance in amnestic MCI and older adults with cognitive complaints. Archives of Clinical Neuropsychology, 23(3), 229–241. doi:10.1016/j.acn.2008.01.005

Comment 5. The "Global cognition" section should appear just before the description of the neuropsychological tests used, as it represents an initial evaluation of the subjects.

Response. The section is now as suggested by Reviewer #3 (see Page 7: line 17)

Comment 6. However, my major concern is about the way the data analyses were performed as they seem redundant at some point. The authors use t-tests (to analyze the results of both groups regardless of genotype) and ANOVAs (to see the effects of group and genotype) and subsequently report group comparisons from t-tests and also the main group effects in the ANOVA. This information is practically the same. In my opinion, it would only be necessary to use the ANOVA to observe these effects. Additionally, I do not quite understand why the authors have "created" 6 groups. Using a 2-way ANOVA can include group (with two levels: young and older) and genotype (three levels: Val/Val, Met/Met, and Met/Val) as factors.

Response. The rationale for our study was first to determine the presence of age-related effects on the three composite scores we created, and next to explore effect of genotypic variations on executive processes that are affected by aging. However, we obtained age-effect on the three scores, and thus the three were included in analyses assessing genetic effects. We have slightly reformulated the end of the introduction to make this point more explicit. Page 5: Lines 14-17: “Second, we tested the COMT allelic variance effects on the executive processes for which an age-effect was found. This was done on the six groups of subjects by combining age (younger, older) and genotype (Met/Met, Val/Met, Val/Val).” Regarding your second comment, as indicated Page 13: Line 3, we decided to include six groups, and not stratifying by age and genotype in order to enter the tree executive scores in the same analysis, and not perform three separate ANOVAs. This was done to decrease the number of statistical tests and risk of false positive. 

Comment 7. I believe it would be advisable to perform the Hardy-Weinberg equilibrium test to analyze if the genotype distributions in the sample are comparable to the general population.

Response. We chose not to include participants directly from the general population until we reached the set N, which meant that we should have tested whether the frequency distribution of the different genotypes in our sample matched that of the population. Instead, we decided to create groups of roughly equivalent size based on a database available in the context of another research protocol (see Reuter et al., 2005 for a similar recruitment protocol). It therefore seems inappropriate to carry out the Hardy-Weinberg equilibrium test to analyze if the genotype distributions in the sample is comparable to the general population. This is now stated Page 6 lines 4-7: “Participants were recruited from a local database at the GIGA-CRC. They were initially enrolled through advertisement in senior centers, universities, and drugstores. Possible subjects offered inclusion criteria presented at the laboratory first for DNA sampling using Isohelix® buccal swabs” and Page 6 Lines 17-18: “The selection of participants from an existing database made it possible to form groups of roughly equivalent size for each genotype (see (61) for a similar procedure).”

61. Reuter, M., Peters, K., Schroeter, K., Koebke, W., Lenardon, D., Bloch, B., & Hennig, J. (2005). The influence of the dopaminergic system on cognitive functioning: A molecular genetic approach. Behavioural and Brain research, 164, 93-99. doi:10.1016/j.bbr.2005.06.002

Comment 8. Finally, the sample size may be considered small for a genetic study; however, the authors could report statistical power or at least conduct sensitivity analyses to observe possible violations of Type II error.

Response. A sensitivity analysis was performed for our ANOVA including the three executive composite scores. Regarding the between effect (Group effect), with a total sample size of 100 and six groups, we were able to detect effect size of f = .30 which corresponds to an η2 = .08 and is an intermediate effect size (d Cohen = 0.60). This is now indicated Page 13 Lines 13-16.

---

## [Decision Letter · Decision Letter 2]

24 Apr 2024

Association between executive functions and COMT Val108/158Met polymorphism among healthy younger and older adults : a preliminary study

PONE-D-23-21633R2

Dear Dr. Collette,

We’re pleased to inform you that your manuscript has been judged scientifically suitable for publication and will be formally accepted for publication once it meets all outstanding technical requirements.

Kind regards,

Kenji Tanigaki, Ph.D., M.D.

Academic Editor

PLOS ONE

Additional Editor Comments (optional):

Reviewers' comments:

Reviewer's Responses to Questions

**Comments to the Author**

1. If the authors have adequately addressed your comments raised in a previous round of review and you feel that this manuscript is now acceptable for publication, you may indicate that here to bypass the “Comments to the Author” section, enter your conflict of interest statement in the “Confidential to Editor” section, and submit your "Accept" recommendation.

Reviewer #3: All comments have been addressed

2. Is the manuscript technically sound, and do the data support the conclusions?

Reviewer #3: Yes

3. Has the statistical analysis been performed appropriately and rigorously? 

Reviewer #3: Yes

4. Have the authors made all data underlying the findings in their manuscript fully available?

Reviewer #3: No

5. Is the manuscript presented in an intelligible fashion and written in standard English?

Reviewer #3: Yes

6. Review Comments to the Author

Reviewer #3: The authors have satisfactorily resolved all my comments. Therefore, I suggest that the article be accepted in its current form.

7. PLOS authors have the option to publish the peer review history of their article (what does this mean?). If published, this will include your full peer review and any attached files.

Reviewer #3: No

---

## [Editor Report · Acceptance letter]

30 Apr 2024

PONE-D-23-21633R2 

PLOS ONE

Dear Dr. Collette, 

I'm pleased to inform you that your manuscript has been deemed suitable for publication in PLOS ONE. Congratulations! Your manuscript is now being handed over to our production team.

Kind regards, 

on behalf of

Dr. Kenji Tanigaki 

Academic Editor

PLOS ONE